# Peripheral Endocannabinoid Components and Lipid Plasma Levels in Patients with Resistant Migraine and Co-Morbid Personality and Psychological Disorders: A Cross-Sectional Study

**DOI:** 10.3390/ijms25031893

**Published:** 2024-02-04

**Authors:** Sara Bottiroli, Rosaria Greco, Valentina Franco, Annamaria Zanaboni, Michela Palmisani, Gloria Vaghi, Grazia Sances, Roberto De Icco, Cristina Tassorelli

**Affiliations:** 1Department of Brain and Behavioral Sciences, University of Pavia, 27100 Pavia, Italy; sara.bottiroli@mondino.it (S.B.); annamaria.zanaboni@mondino.it (A.Z.); gloria.vaghi@mondino.it (G.V.); roberto.deicco@unipv.it (R.D.I.); cristina.tassorelli@unipv.it (C.T.); 2Headache Science and Neurorehabilitation Center, IRCCS Mondino Foundation, 27100 Pavia, Italy; valentina.franco@mondino.it (V.F.); michela.palmisani@mondino.it (M.P.); grazia.sances@mondino.it (G.S.); 3Clinical and Experimental Pharmacology Unit, Department of Internal Medicine and Therapeutics, University of Pavia, 27100 Pavia, Italy

**Keywords:** resistant migraine, endocannabinoid system, personality disorders, psychological comorbidities, depression

## Abstract

Resistant migraine characterizes those patients who have failed at least three classes of migraine prophylaxis. These difficult-to-treat patients are likely to be characterized by a high prevalence of psychological disturbances. A dysfunction of the endocannabinoid system (ECS), including alteration in the levels of endocannabinoid congeners, may underlie several psychiatric disorders and the pathogenesis of migraines. Here we explored whether the peripheral gene expression of major components of the ECS and the plasma levels of endocannabinoids and related lipids are associated with psychological disorders in resistant migraine. Fifty-one patients (age = 46.0 ± 11.7) with resistant migraine received a comprehensive psychological evaluation according to the DSM-5 criteria. Among the patients, 61% had personality disorders (PD) and 61% had mood disorders (MD). Several associations were found between these psychological disorders and peripheral ECS alterations. Lower plasma levels of palmitoiletanolamide (PEA) were found in the PD group compared with the non-PD group. The MD group was characterized by lower mRNA levels of diacylglycerol lipase α (*DAGL*α) and *CB2* (cannabinoid-2) receptor. The results suggest the existence of peripheral dysfunction in some components of the ECS and an alteration in plasma levels of PEA in patients with resistant migraine and mood or personality disorders.

## 1. Introduction

In recent years, many advances have been reached in the field of migraine preventive treatment [1]. Unfortunately, not all patients achieve the same benefits [2]. The consensus statement from the European Headache Federation (EHF) [3] defined two particularly aggressive forms of the disease in which the patient does not benefit from any of the preventive therapies: resistant and refractory migraines. Resistant migraine is characterized by the failure of at least three classes of preventive drugs, whereas refractory migraine consists of the failure of all available preventative drugs, including monoclonal antibodies (mABs) targeting the calcitonin gene-related peptide (CGRP). From a psychological viewpoint, these difficult-to-treat patients are frequently characterized by a high prevalence of psychological vulnerabilities [4,5,6,7], having an impact on the migraine treatment outcome [8,9,10]. Consequently, the investigation of psychological disorders in patients with difficult-to-treat migraines becomes very important, as they might influence the course of the disease itself and the response to treatment [11]. We recently demonstrated that psychological disorders represent predictors of negative outcomes of mABs treatment in chronic migraine (CM) [9,12]. Hence, even if mABs block the CGRP pathway and attenuate peripheral/central sensitization, their effect is attenuated by the simultaneous role of psychiatric comorbidities in the opposite direction. Taken together, these findings are interesting in order to try to shed light on those psychological factors involved in the physiology of migraines. The biopsychosocial model of health [13,14] postulates the existence of a complex interaction between psychological, psychosocial, and biological aspects, reciprocally influencing each other. Consequently, what is critical is the identification of biomarkers being involved in both psychological disorders and migraines.

In recent years, it has been shown that among the neurobiological systems involved in mental disorders, the endocannabinoid system (ECS) appears to play an active role [15]. Dysregulation of the various ECS components may have a role in the pathophysiology of mental diseases [16,17,18], such as schizophrenia, anxiety-related disorders, and depression [15]. The majority of studies here have focused on cannabinoid-1 (CB1) receptor, as well as on fatty acid amide hydrolase (FAAH), in cerebral regions connected to the amygdala–hippocampal–cortico–striatal neural circuit [15]. CB1 is abundantly expressed in the brain, while cannabinoid-2 (CB2) receptor is mainly expressed in peripheral tissues [19]. CB1 receptor gene variants increase the risk of migraines [20,21] and are associated with the presence of depression and anxiety in a large general population sample [22]. Furthermore, genetically reduced FAAH activity and repetitive stress in childhood are associated with increased susceptibility to anxiety and depression in later life [23].

Growing evidence suggests that the ECS also has a role in migraine pathophysiology and may modulate disease-related pain circuits [24,25]. For instance, anandamide (AEA) appears to be markedly reduced in the cerebrospinal fluid of CM [26]. Although more evident in the chronic subtype, the peripheral gene expression of enzymes involved in AEA and 2-acylglycerol (2-AG) metabolism has also been reported to be altered in subjects with episodic migraine [27]. Thus, the existing literature suggests that dysfunctions of the ECS may be involved in the origin of both psychological disorders and migraine chronification.

The present study aimed to evaluate ECS components’ dysfunction in resistant migraine when associated with psychological disturbances by considering possible differences in *CB1* receptor expression as the primary outcome. Specifically, to provide insights into the neurobiological mechanisms, we examined ECS components’ gene expression in peripheral blood mononuclear cells (PBMCs), as well as the content of endocannabinoids and related lipids in plasma.

## 2. Results

### 2.1. Patient Population

We recruited 51 patients (88% females; mean age 46.0 ± 11.7; age range 22–65). Clinical and psychological profiles of the study population are reported in Table 1. Twenty-five percent of participants (n = 13) were taking antidepressants at the time of enrollment. At the psychological evaluation, 61% of participants satisfied the criteria for personality disorders (PD), mostly belonging to Cluster C and to obsessive compulsive PD, 61% presented mood disorders (MD), and 43% satisfied the criteria for comorbid PD and MD. The distribution of patients in the PD/wPD and MD/wMD groups is reported in Appendix A. We also found a high prevalence of patients presenting anxiety disorders (82%), in line with existing studies.

### 2.2. Comparison between Subjects with PD (PD) and without PD (wPD)

When comparing demographic, clinical, and psychological characteristics between the PD and the wPD groups, as reported in Table 2, we did not detect significant differences. Both groups had a high prevalence of anxiety and mood disorders.

As regards the ECS components assessed in the PBMCs, as reported in Figure 1a,b, no significant difference in gene expression of ECS components was observed between PD and wPD. In terms of plasma lipid levels, the PD group had lower levels of palmitoiletanolamide (PEA) (*p* = 0.038) than the wPD group. The levels of PEA were not associated with sex, age, ongoing antidepressive treatment, or migraine days per month according to the robust regression model. No significant difference was reported in the AEA plasma levels (*p* = 0.078). 2-AG plasma concentrations were below the detection level in all patients (Figure 1c).

### 2.3. Comparison between Subjects with MD and Subjects without MD (wMD)

When comparing demographic, clinical, and psychological characteristics between MD and wMD, as reported in Table 3, we detected significant differences in terms of anxious (*p* = 0.032) and depressive (*p* = 0.006) symptomatology, as resulted from the HADS and the SF-36 mental subscale (*p* = 0.004), in all cases unfavorable to the MD group. In addition, the MD group was characterized by a higher prevalence of patients with anxiety disorders than the wMD group (χ2 (1, n = 51) = 6.82, *p* = 0.02), suggesting that it was a more complicated pool of participants from a psychological point of view.

When assessing gene expression levels in the PBMCs, as shown in Figure 2a,b, the MD group was characterized by lower levels of CB2 (*p* = 0.031) and diacylglycerol lipase α (DAGL) (*p* = 0.033) gene expression than the wMD group. These levels of CB2 and DAGL were not associated with sex, age, ongoing antidepressive treatment, or migraine days per month according to the robust regression model. No other significant differences in the gene expression of ECS components were observed between these two groups, mor were their differences in terms of plasma lipid levels (Figure 2c).

## 3. Discussion

Resistant migraine is a condition frequently encountered in clinical practice, representing a challenge for clinicians [28]. From a clinical point of view, these patients are considered difficult to treat, given the lack of response to at least three classes of migraine preventatives [3]. From a psychological point of view, the outcome of treatment, including pharmacological therapies, is often associated with the complexity of the patient’s psychological profile [6,8,9,10]. This means that the more psychological disturbances (e.g., anxiety, depression, personality disorders, compromised socio-cognitive abilities, stress, and life events) the patient has, the more likely treatment failure is. Several hypotheses could explain this association according to the biopsychosocial model of health [13,14], but the underlying biological mechanisms have yet to be explored. Dysfunction of the ECS may indeed underlie several psychiatric disorders [29,30,31] and the chronification of migraines [27,32,33]. In particular, the ECS is a crucial component of several brain circuits, including those that control pain perception [34], stress, and emotion regulation [35].

As difficult-to-treat patients, it is not surprising that more than half of our study population presented with personality disorders and/or major depression. These results are consistent with the prevalence of psychological disorders that we previously observed [9] in CM patients and highlight the complication from the psychological point of view that characterizes patients with multiple treatment failures [5,6,8]. Most personality disorders belong to Cluster C, which reflects an “anxious-fearful” and stress-reactive personality [36]. In particular, in line with previous studies on CM [37,38,39], obsessive compulsive PD was the most prevalent Cluster C subtype among our study population. The prototypical description of the “obsessive compulsive personality” refers to individuals who tend to adhere rigidly to their daily routine, becoming uncomfortable and anxious when something goes wrong [40,41]. Interestingly, 43% of our participants had a comorbidity of personality and mood disorders. This is consistent with previous studies [42,43] that have shown recurrent co-occurrence between depressive disturbances and personality disorders and that, in particular, Cluster C diagnoses are particularly common among depressed patients [42,44,45]. In the context of the present study population, these findings could be explained by the fact that Cluster C and obsessive-compulsive PD—because of their rigidity characteristics—can easily lead to depression as a result of the exposure to uncontrollable events, such as recurrent migraine.

The current study’s attempt to link personality and mood problems with gene expression of some components of the ECS in PBMCs and with plasma levels of endocannabinoids and lipids in patients with resistant migraine is novel and intriguing. To the best of our knowledge, this is the first study to analyze changes in the ECS in migraine subjects according to their psychological comorbidities. Previous studies reported that multiple *CB1* gene polymorphisms have been observed in depressed patients [46], some of which are associated with lower CB1 receptor expression in specific brain areas [47,48]. Furthermore, resistance to treatment was observed in patients with depression having a single nucleotide polymorphism in the *CB1* receptor [49]. Animals with low levels of endocannabinoid signaling show anxiety, depression and schizophrenia-like behavior representative of psychiatric disorders observed in patients [15,50]. Changes in ECS components were also detected in PBMCs in other neurological diseases [51,52]. For instance, schizophrenic patients showed elevated AEA plasma levels associated with reduced *CB1* gene expression in PBMCs [53,54], and a positive correlation between *CB1* receptor gene expression in monocytes and cognitive impairment [55].

Here, the gene expression of ECS components and PPAR receptors in PBMCs was not significantly different between PD and wPD. Surprisingly, we found significantly reduced plasma levels of PEA in PD compared with wPD, but no significant change in plasma levels of AEA and N-oleoylethanolamide (OEA) was observed. The observed reductions in plasma PEA could be related to the increased inflammatory state found in the PD patients. PEA action can indirectly involve transient potential receptor vanilloid receptor type 1 (TRPV1) channels and G protein-coupled receptor 55 (GPR55) [56], a non-CB1/CB2 cannabinoid receptor, involved in behavioral, immunological, and neuroinflammatory functions. Thus, additional studies are needed to better understand the association between ECS and psychological alterations.

Our findings are coherent with those suggesting a role for PEA as a potential biomarker in stress and depression [57], extending it to the field of PDs. PEA regulates synaptic plasticity, neurogenesis, and monoaminergic neurotransmission—all crucial processes dysregulated in depressive disorder and related dysfunctions [58,59]. Similarly, a significant reduction in the plasma levels of OEA, a shorter monounsaturated analogue of the AEA, has been reported in patients with post-traumatic stress disorder and severe post-traumatic stress disorder symptoms [60].

Furthermore, consider that most of the PD disorders belonged to Cluster C, being characterized by stress-reactive and control-maladaptive personalities. Thus, this inference could be further supported by the fact that difficult-to-treat migraine patients probably experienced numerous physiological and behavioral changes caused by chronic stress, including social deficits, decreased endocrine function, and an increased chance of developing psychiatric illness [61].

The involvement of ECS and related lipids in numerous physiological and pathophysiological processes, such as the control of emotional behavior, cognitive function, inflammation, chronic pain, epilepsy, and, in general, their role in underlying neuropsychiatric disorders have attracted much attention [16]. Thus, as secondary outcomes, we considered ECS alterations in mood. Here, we found that MD in resistant migraine was associated with lower mRNA levels of *CB2* and *DAGL*α in peripheral cells than wMD. However, the decreased *CB2* and *DAGL* gene expression was not associated with changes in 2-AG plasma levels, which were below the threshold for detention. In agreement with this, in patients with psychosis, the protein expression of CB2 receptors was significantly downregulated along with the gene expression of *DAGL*α in PBMCs [18]. In addition, genetic deletion of the enzyme DAGL in mice reduces the brain, but not circulating, levels of 2-AG, causing sex-specific anxious and anhedonic phenotypes associated with altered endocannabinoid signaling [62]. CB2 receptor is highly expressed in the leukocyte subpopulation in humans [63] and regulates the immune system [64]. The CB2 receptors are potential immunomodulatory with specific roles in cell-type specificity and they are localized in the brain regions involved in emotional behavior and stress coping throughout the central nervous system [65,66,67]. Indeed, emotional behavior has been associated with immune system alterations in psychological diseases [68,69,70]. The gene expression of *CB2* receptor was upregulated in PBMCs [71] and cells of the innate immune system [72] and was correlated with positive and negative syndrome scale and cognitive performance severity [54,71].

The present study raises certain unresolved inquiries that urge a careful approach when interpreting the findings, prompting the need for additional investigations. One crucial query revolves around the origin of PEA levels. There is evidence that this lipid may also be produced in the vascular endothelium, and blood cells are a significant circulating source of ECS and related lipids. Patients with psychopathologies present alterations compatible with moderate endothelial dysfunction [73,74,75,76,77]. For instance, it has been shown that specific personality traits, such as neuroticism and alexithymia, that are particularly prevalent in the migraine population [6,78], are associated with endothelial dysfunction [79]. Moreover, it should be considered that chronic stress impairs endothelium function via the release of stress hormones such as glucocorticoids, pro-inflammatory cytokines, and endothelin-1 in reaction to mental stress [80]. Regarding this point, it should be noted that 82% of patients in this study were diagnosed with anxiety disorders, which could be related to the presence of intense stress [81,82]. Ultimately, it remains uncertain as to whether diminished mRNA levels of *CB2* and *DAGL* translate into operational receptors and enzymes, respectively. For instance, a decline in *CB2* gene expression might suggest a decrease in quantity rather than an inherent change in function. Conducting additional experiments to investigate the functional capabilities of these receptors and enzymes will contribute to a better comprehension of the data’s significance.

### Limitations of the Study

Some limitations within our present study call for caution in result interpretation and underscore the need for further research. To begin with, a significant majority (88%) of the study’s participants were female, which does correspond to the epidemiology of chronic and episodic high-frequency migraines. However, the relatively small number of male subjects restricts the generalizability of our findings to the male population. Furthermore, it should be noted that we found a 43% comorbidity rate between depression and personality disorders. This implies that the majority of our patients in both the PD and MD groups exhibited comorbidities. While this aspect is well documented in the literature [83,84], it may have influenced our results, as we did not have distinct patient groups for the respective psychological disorders as well as a control group of treatable migraine patients. Finally, although our statistical calculations were accurate, the sample size remained relatively small, which could potentially limit the scope of our result interpretations. For all these reasons, future investigations, encompassing larger cohorts of migraine sufferers, are essential to corroborate our findings.

## 4. Materials and Methods

### 4.1. Resistant Migraine Patients

The inclusion criteria were as follows: (a) age > 18, <65 years; (b) fulfillment of EHF consensus criteria for resistant migraine [3]; (c) at least 8 days of migraine/month; (d) Migraine Disability Assessment Questionnaire (MIDAS) score ≥ 11 at enrollment; and (e) previous failure of at least three different pharmacological classes of preventive therapies. The exclusion criteria were as follows: (a) dementia, (b) previous diagnosis of psychosis and (c) intellectual disabilities. A previous therapeutic failure was defined as follows: (a) no reduction (<30%) in headache frequency after at least 6 weeks of treatment with an adequate dose or (b) the subject discontinued the treatment due to related adverse events or poor tolerability. An expert neurologist verified the eligibility criteria during the recruitment process based on history, headache diaries, and neurological evaluation.

### 4.2. Procedure

The data presented here represent the baseline evaluation of patients who were enrolled in monocentric open label study aimed at identifying predictors of response to CGRP-targeting mABs in migraine patients. The study started in 2022 at the Headache Science and Neurorehabilitation Center (a tertiary referral center) of the Mondino Foundation in Pavia, Italy and was approved by the local Ethics Committee (# P-20210047311) and registered at https://clinicaltrials.gov (NCT05046119) (accessed on 18 December 2023).

The patients fulfilling criteria for inclusion in the study were enrolled and signed the informed consent form at the screening visit (visit 0). They were instructed to record headache characteristics in an ad hoc diary for one month (run-in period) and they returned to the center to receive the first dose of mAB (Visit 1). During visit 1, patients underwent a thorough clinical and psychological profiling, venous puncture for the collection of a blood sample (30 mL) and were treated with one of the three mABs targeting CGRP available on the Italian market. A follow-up visit was scheduled three months later (visit 2) to assess response to treatment.

### 4.3. Psychological Evaluation

The psychological evaluation was performed by an expert psychologist (SB) based on the Diagnostic and Statistical Manual of Mental Disorders (DSM-5) criteria [85] using the Structured Clinical Interview for DSM-5, Clinical Version (SCID-5-CV) [86] for assessing personality disorders, mood, and anxiety disturbances. Interview questions were provided alongside each DSM-5 criterion to aid users in rating each criterion as either present or absent. Personality disorders comprise 10 disorders, which can be grouped into Cluster A (paranoid, schizoid, and schizotypal), Cluster B (antisocial, borderline, histrionic, and narcissistic), and Cluster C (avoidant, borderline, and dependent) according to the shared characteristics. Anxiety disorders include specific phobias, social anxiety disorder, and generalized anxiety disorder, as well as panic disorder and agoraphobia. Mood disorders include bipolar disorder, cyclothymia, major depressive disorder, disruptive mood dysregulation disorder, persistent depressive disorder, and premenstrual dysphoric disorder.

Participants also filled out a series of questionnaires. The Leeds Dependence Questionnaire (LDQ) [87] and the Severity of Dependence Scale (SDS) [88,89] were used to assess dependence on acute medications for migraine. The Italian version of the Hospital Anxiety and Depression Scale (HADS) [90] was used to assess anxiety and depression symptomatology. This questionnaire comprises seven items concerning depression and seven items for anxiety, graded on a four-point (0–3) Likert scale, so that possible scores range from 0 to 21 for both depression and anxiety. The 36-Item Short Form Health Survey questionnaire (SF-36) [91] was used in order to assess the quality of life. It measures eight domains of health status: physical functioning (10 items); physical role limitations (four items); bodily pain (two items); general health perceptions (five items); energy/vitality (four items); social functioning (two items); emotional role limitations (three items); and mental health (five items). A scoring algorithm was used to convert the raw scores into the mental and physical sub-dimensions.

### 4.4. Isolation of Peripheral Blood Mononuclear Cells (PBMCs)

Blood samples (20 mL) were collected within ethylenediamine tetra-acetic acid (EDTA) containing tubes from participants (interictal period, between 8:00 a.m. and 12:00 a.m.) and diluted at a 1:1 ratio with phosphate-buffered saline 1X (PBS 1X) (Sigma, Milan, Italy). Diluted blood samples were slowly loaded onto Ficoll separating solution (15 mL) (Sigma) and centrifuged at 800× *g* without brake for 30 min at room temperature. PBMCs accumulated as the middle white monolayer were washed twice in sterile PBS 1X and centrifuged at 300× *g* for 15 min. The PBMCs were used for RNA extraction for the evaluation of the gene expression of CB receptors, FAAH and monoacylglycerol lipase (MAGL) (the enzymes that degrade the endocannabinoids AEA and 2-acylglycerol (2-AG), respectively), along with N-acyl-phosphatidylethanolamine (NAPE)-hydrolyzing phospholipase D (NAPE-PLD) and diacylglycerol lipase α (DAGLα) (the substrates of AEA and 2-AG, respectively).

### 4.5. Gene Expression of ES Components

Following the manufacturer’s instructions, total RNA from PBMCs was isolated using the standard methods (Zymo Research, Roma, Italy), and the quality of the RNA was evaluated using a NanoDrop spectrophotometer (NanoDrop Technologies, Thermofisher, Waltham, MA, USA). cDNA was then produced using the iScript cDNA Synthesis kit (Bio-Rad, Milan, Italy). Using the Fast Eva Green supermix, we assayed the gene expression of CB receptors, FAAH, MAGL, NAPE-PLD and DAGLα (Bio-Rad).

Table 4 lists the primer sequences that were acquired using the AutoPrime program http://www.autoprime.de/AutoPrimeWeb (accessed on 18 December 2023). As a housekeeping gene, ubiquitin C (UBC), whose expression was constant across all experimental groups, was employed. Following the supplier’s instructions, the amplification was carried out using a light Cycler 480 Instrument rt-PCR Detection System (Roche). All samples were assayed in triplicate and gene expression levels were calculated according to the 2^−ΔΔCt^ = 2−(ΔCt gene − ΔCt housekeeping gene) formula by using Ct (cycle threshold) values.

### 4.6. Quantitative Profiling of AEA, 2-AG and Related Lipids by LC-MS/MS

For the measurement of AEA, 2-AG, palmitoylethanolamide (PEA) and N-oleoylethanolamide (N-OEA), 100 μL of plasma was mixed with 10 μL of the internal standard cannabidiol-d3 solution (5000 ng/mL in methanol with 0.8% formic acid), 10 μL of methanol with 0.8% formic acid, and 880 μL of methanol. After 10 min of centrifugation (17,000× *g* at 4 °C), the supernatant was transferred to a glass clean tube and evaporated to dryness under a gentle stream of nitrogen at room temperature. The dry residue was reconstituted with 48 μL of methanol with 0.8% formic acid and 12 μL of water, vortex mixed and transferred to a polypropylene vial to be injected in an online solid phase extraction technique coupled with liquid chromatography-electrospray tandem mass spectrometry according to Fanelli et al. with modifications [92].

Analyses were performed using an ExionLC 100 system (Applied Biosystems Sciex, Darmstadt, Germany) and a 3200 QTRAP^®^ triple quadrupole mass spectrometer (Applied Biosystems Sciex, Darmstadt, Germany). An online solid phase extraction procedure was implemented with a perfusion column (POROS R1, 2.1 × 30 mm i.d., 20 μm, Thermo Fisher Scientific, Waltham, Massachusetts, United States). A monolithic Onyx C18 column (100 mm × 3 mm i.d., Phenomenex, Bologna, Italy) kept at 25 °C was employed to achieve an enhanced analytical separation. Mobile phase A consisted of 10 mM ammonium formate and 0.1% formic acid in water/methanol (98:2 *v*/*v*), mobile phase B was 8 mM ammonium formate and 0.08% formic acid in methanol/acetonitrile/isopropanol (80:10:10 *v*/*v*).

### 4.7. Primary and Secondary Outcome Measures

The primary outcome measure of the study was the difference in CB1 receptor expression between subjects with/without personality disorders (PD). The association between other measures of the ECS and PD or mood disorders (MD) were evaluated as exploratory secondary outcome measures. Anxiety disorders were considered only for describing patients and not for their classification, given the expected high prevalence among enrolled patients.

### 4.8. Statistical Procedures

The sample size was calculated based on the primary outcome measure. Considering an alpha significance of 0.05, a test power of 0.8, and an effect size of 0.1, the minimum sample size needed to detect a significant correlation between PD and CB1 receptor abnormalities was 17 subjects in each group (with/without PD) for a total of 34 subjects. Since the proportion of PD patients cannot be determined at enrolment, a sample size of about 50 subjects was expected to be enrolled. The sample size was calculated using G*Power Version 3.1.9.4 software.

Data for ECS components of interest were presented as the median and 5th–95th percentiles, whereas demographic, clinical, and psychological characteristics of enrolled patients as the mean ± standard deviation (continuous variables) and n/% (categorical variables). Patients were classified according to the presence or absence of psychological disorders, i.e., PD and MD. Considering the non-normal distribution of the collected measures as resulted from Q-Q plots, we used non-parametric tests for comparing groups. We used the Mann–Whitney U test (continuous features) and Fisher’s exact test (categorical features). Analysis focused on comparing (1) PD vs. without PD (wPD) and (2) MD vs. wMD. To reduce the impact of outliers in the multivariate analysis, a robust regression model was implemented for markers shown to be significantly different in the Mann–Whitney test. The software performs a regression, calculates case weights from absolute residuals, and regresses again using these weights. The significance level threshold was set at 0.05. All analyses were conducted using STATA v(17.0).

## 5. Conclusions

The results of the present study expand on the factors that may be involved in the pathophysiology of resistant migraine, especially when associated with personality and/or mood disorders, and are essential to further differentiate this complex group of difficult-to-treat patients into different phenotypic and/or endotypic subtypes. Most studies to date have assessed ECS and related lipids in migraine sufferers without making any distinctions based on psychological comorbidities identified by a rigorous in-person assessment. Our study’s attempt to shed light on this issue is novel and intriguing and is extremely important because it sheds light on an association that has rarely been explored so far and opens the way to the development of new therapeutic approaches for a particularly challenging migraine population.

## Figures and Tables

**Figure 1 ijms-25-01893-f001:**
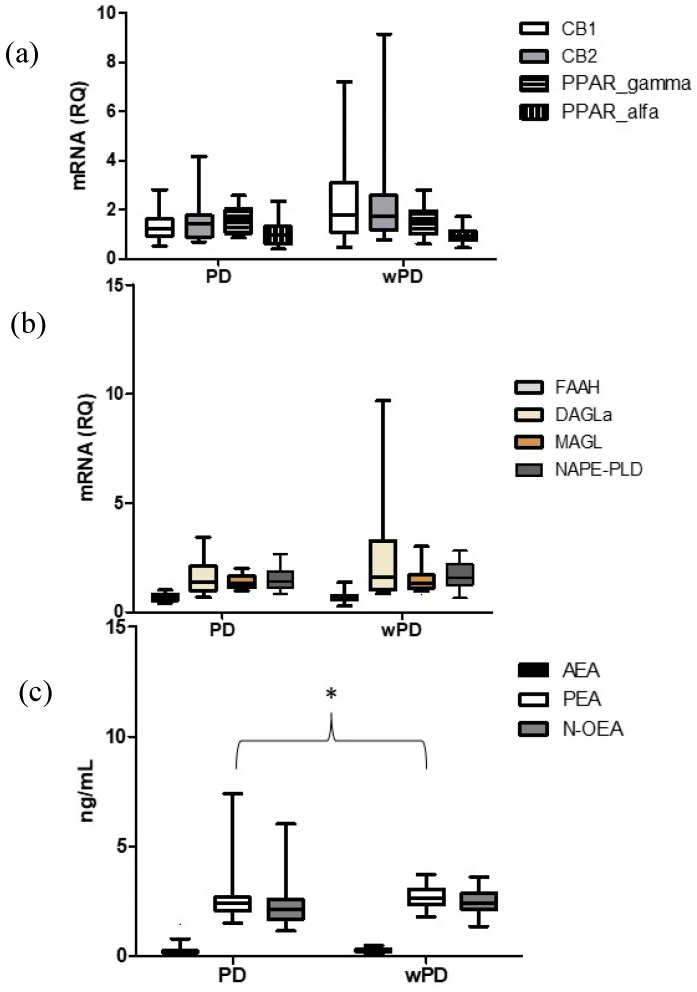
Comparison of the mRNA levels of endocannabinoid system components in PBMCs (**a**,**b**) and plasma lipid levels (**c**) in patients with personality disorder (PD) and without personality disorder (wPD). Legend. RQ = Relative quantification: 2^−∆∆Ct^ = 2 − (∆Ct gene − ∆Ct housekeeping gene), Ct = cycle threshold. CB1 = cannabinoid receptor type 1; CB2 = cannabinoid receptor type 2; PPAR = peroxisome proliferator-activated receptors; FAAH = fatty acid amide hydrolase; DAGL α= diacylglycerol lipases α; MAGL = monoacylglycerol lipase; NAPE-PLD = N-Acyl-phosphatidylethanolamine-hydrolyzing phospholipase D; AEA = anandamide; PEA = palmitoylethanolamide; N-OEA = N-oleoylethanolamide. * denotes significant differences.

**Figure 2 ijms-25-01893-f002:**
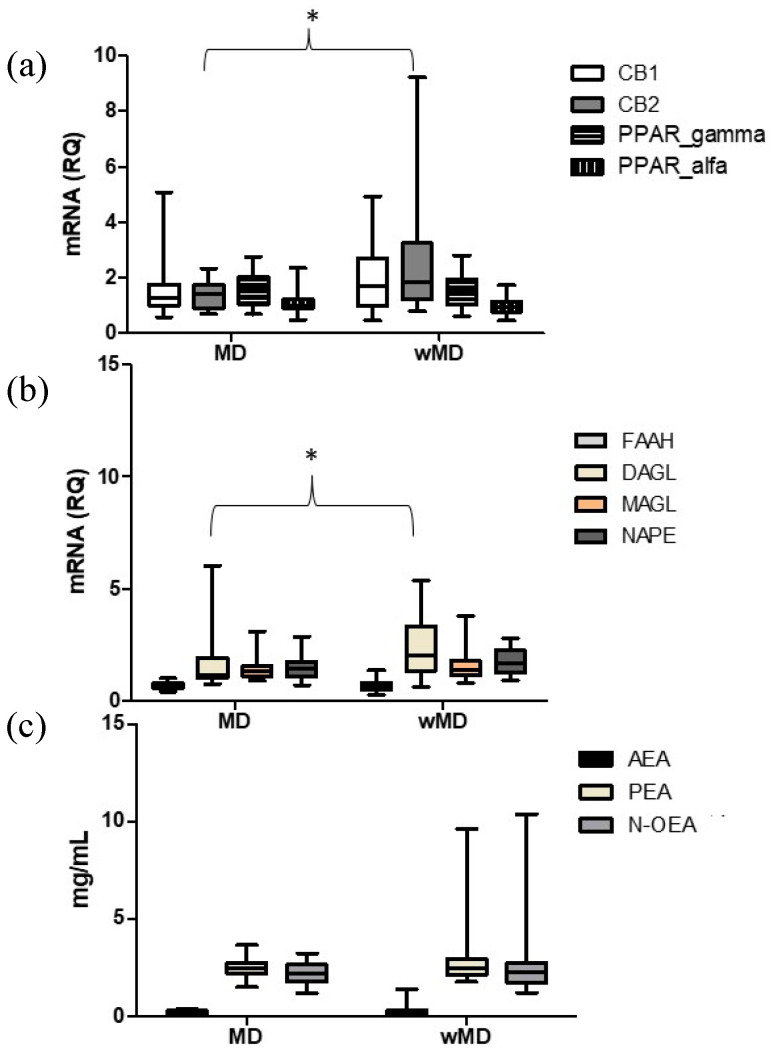
Comparison of the mRNA levels of endocannabinoid system components in PBMCs (**a**,**b**) and plasma lipid levels (**c**) in patients with a mood disorder (MD) and without a mood disorder (wMD). Legend. RQ = Relative quantification: 2^−∆∆Ct^ = 2 − (∆Ct gene − ∆Ct housekeeping gene), Ct = cycle threshold. CB1 = cannabinoid receptor type 1; CB2 = cannabinoid receptor type 2; PPAR = peroxisome proliferator-activated receptors; FAAH = fatty acid amide hydrolase; DAGL α= diacylglycerol lipases α; MAGL = monoacylglycerol lipase; NAPE-PLD = N-Acyl-phosphatidylethanolamine-hydrolyzing phospholipase D; AEA = anandamide; PEA = palmitoylethanolamide; N-OEA = N-oleoylethanolamide. * denotes significant differences.

**Table 1 ijms-25-01893-t001:** Demographic, clinical, and psychological characteristics of study population at enrolment. Data are presented as “mean ± standard deviation” or “absolute value.

	Total n = 51
Age	46.0 ± 11.7
Gender, female	45 (88%)
CM	38 (75%)
Ongoing antidepressants treatment	13 (25%)
Migraine days per month	20.2 ± 7.3
Days of acute drug intake per month	16.5 ± 6.9
Doses of acute drug intake per month	24.1 ± 15.7
MIDAS	79.2 ± 38.1
HIT-6	67.6 ± 4.3
LDQ	7.3 ± 4.4
SDS	6.9 ± 3.1
HADS anxiety subscale	5.7 ± 3.8
HADS depression subscale	5.9 ± 4.2
SF-36 physical subscale	35.4 ± 7.6
SF-36 mental subscale	39.6 ± 10.9
**Personality disorders**	31 (61%)
Cluster A	3 (6%)
Paranoid	3 (6%)
Schizoid	0 (0%)
Schizotypal	0 (0%)
Cluster B	7 (14%)
Histrionic	3 (6%)
Narcissistic	5 (10%)
Antisocial	0 (0%)
Borderline	2 (4%)
Cluster C	31 (61%)
Avoidant	3 (6%)
Dependent	1 (2%)
Obsessive compulsive	31 (61%)
Anxiety disorders	42 (82%)
Mood disorders	31 (61%)
Comorbid personality and mood disorders	22 (43%)

Note. CM = chronic migraine; MIDAS = Migraine Disability Assessment; HIT-6 = The Headache Impact Test; LDQ = Leeds Dependence Questionnaire; SDS = Severity of Dependence Scale; HADS = Hospital Anxiety and Depression Scale; SF-36 = 36-Item Short Form Health Survey questionnaire. Cluster A includes paranoid, schizoid, and schizotypal personality disorders; Cluster B includes antisocial, borderline, histrionic, and narcissistic personality disorders; Cluster C includes avoidant, borderline, and dependent personality disorders.

**Table 2 ijms-25-01893-t002:** Demographic, clinical, and psychological characteristics of PD and wPD groups at enrolment. Data are presented as “mean ± standard deviation” or “absolute value.

	PD n = 31	wPD n = 20	*p*
Age	46.4 ± 11.8	45.5 ± 11.8	0.80
Gender, female	28 (90%)	17 (85%)	0.44
CM	21 (68%)	17 (85%)	0.20
Ongoing antidepressants treatment	11 (36%)	2 (15%)	0.04
Migraine days per month	20.5 ± 7.5	20.0 ± 7.2	0.90
Days of acute drug intake per month	15.6 ± 6.1	17.8 ± 7.9	0.44
Doses of acute drug intake per month	23.0 ± 16.0	25.7 ± 15.6	0.30
MIDAS	86.3 ± 43.5	68.2 ± 24.9	0.12
HIT	67.5 ± 5.2	67.6 ± 2.4	0.67
LDQ	7.4 ± 4.4	7.2 ± 4.5	0.76
SDS	6.4 ± 2.7	7.7 ± 3.4	0.29
HADS anxiety subscale	5.7 ± 3.8	5.6 ± 3.9	0.88
HADS depression subscale	5.6 ± 3.8	6.5 ± 4.9	0.54
SF-36 physical subscale	35.5 ± 7.9	35.3 ± 7.4	0.65
SF-36 mental subscale	38.8 ± 10.6	40.9 ± 11.4	0.47
Anxiety disorders	26 (84%)	16 (80%)	0.72
Mood disorders	21 (68%)	10 (50%)	0.25

Note. PD = personality disorder; wPD = without personality disorder; CM = chronic migraine; MIDAS = Migraine Disability Assessment; HIT-6 = The Headache Impact Test; LDQ = Leeds Dependence Questionnaire; SDS = Severity of Dependence Scale; HADS = Hospital Anxiety and Depression Scale; SF-36 = 36-Item Short Form Health Survey questionnaire.

**Table 3 ijms-25-01893-t003:** Demographic, clinical, and psychological characteristics of MD and wMD groups at enrolment. Data are presented as “mean ± standard deviation” or “absolute value”.

	MD n = 31	wMD n = 20	*p*
Age	45.8 ± 11.3	46.4 ± 12.6	0.88
Gender, female	29 (94%)	16 (80%)	0.15
CM	23 (74%)	15 (75%)	0.61
Ongoing antidepressants treatment	11 (36%)	2 (15%)	0.04
Migraine days per month	20.3 ± 6.9	20.2 ± 7.9	0.91
Days of acute drug intake per month	16.1 ± 6.4	17.1 ± 7.7	0.96
Doses of acute drug intake per month	23.3 ± 13.5	25.3 ± 18.9	0.74
MIDAS	84.3 ± 39.4	71.4 ± 35.5	0.21
HIT	68.1 ± 4.8	66.7 ± 3.4	0.07
LDQ	8.2 ± 4.5	6.1 ± 4.1	0.07
SDS	7.3 ± 3.3	6.4 ± 2.7	0.29
HADS anxiety subscale	6.6 ± 3.9	4.2 ± 3.3	0.03
HADS depression subscale	7.3 ± 4.6	3.9 ± 2.6	0.006
SF-36 physical subscale	34.5 ± 6.7	36.8 ± 8.9	0.15
SF-36 mental subscale	36.2 ± 10.4	45.0 ± 9.6	0.004
Anxiety disorders	29 (94%)	13 (65%)	0.02
Personality disorders	21 (68%)	10 (50%)	0.25

Note. MD = mood disorder; wMD = without mood disorder; CM = chronic migraine; MIDAS = Migraine Disability Assessment; HIT-6 = The Headache Impact Test; LDQ = Leeds Dependence Questionnaire; SDS = Severity of Dependence Scale; HADS = Hospital Anxiety and Depression Scale; SF-36 = 36-Item Short Form Health Survey questionnaire.

**Table 4 ijms-25-01893-t004:** Primer sequences of PBMCs.

Gene	Forward Primer	Reverse Primer
*Ubiquitin C*	AGAGGCTGATCTTTGCTGGA	GGGTGGACTCTTTCTGGATG
*CB1*	CCTTTTGCTGCCTAAATCCAC	CCACTGCTCAAACATCTGAC
*CB2*	CATGGAGGAAGGAATGTGCTGGGTGAC	GAGGAAGGCGATGAACAGGAG
*PPARγ*	GGCTTCATGACAAGGGAGTTTC	AACTCAAACTTGGGCTCCATAAAG
*PPARα*	ATGGTGGACACGGAAAGCC	CGATGGATTGCGAAATCTCTTGG
*FAAH*	GAGGACATGTTCCGCTTGGA	TGTTGTCTTGGCAAGAAGGGA
*NAPE-PLD*	TTGTGAATCCGTGGCCAACATGG	TACTGCGATGGTGAAGCACG
*MAGL*	CAAGGCCCTCATCTTTGTGT	ACGTGGAAGTCAGACACTAC
*DAGLα*	AGAATGTCACCCTCGGAATGG	GTGGCTCTCAGCTTCGACAAAGG

*Note*. *CB1* = cannabinoid receptor type 1; *CB2* = cannabinoid receptor type 2; *PPAR* = peroxisome proliferator-activated receptors; *FAAH* = fatty acid amide hydrolase; *NAPE-PLD* = N-Acyl-phosphatidylethanolamine-hydrolyzing phospholipase D; *MAGL* = monoacylglycerol lipase; *DAGLα* = diacylglycerol lipases α.

## Data Availability

The datasets presented in this study can be found in online repositories. The names of the repository/repositories and accession number(s) can be found below: [Zenodo; Reservation: 10.5281/zenodo.7781213].

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
