# Peer review of "Peripheral Endocannabinoid Components and Lipid Plasma Levels in Patients with Resistant Migraine and Co-Morbid Personality and Psychological Disorders: A Cross-Sectional Study"

_ijms, 2024, doi:10.3390/ijms25031893_

Round 1

Reviewer 1 Report

Comments and Suggestions for Authors

Comments to the authors

Thank you for inviting me to review the manuscript entitled “Peripheral endocannabinoid components and lipid plasma levels in patients with resistant migraine and co-morbid personality and psychopathological disorders: A cross-sectional study”. 

This is a cross sectional study conducted on patients with resistant migraine. Briefly, the authors characterized this cohort according to their psychological disorders and quantified the plasma levels of PEA, DAGLalfa and CB receptor gene expression. 

The study has several strengths below identified. I could only point at one major shortcoming, which can be easily acknowledge in the limitation section. Also, other minor corrections are below highlighted. 

Abstract:

-              Line 19 -> “psychopathologies” would be better replaced by “psychological disorders”. Also, it would be better to be consistent throughout the abstract and the manuscript. For example, in line 24, these conditions are named as “psychopathological disorders”, while in line 21 they are referred to as “psychiatric disorders”. 

-              Line 28: PEA and CB1 as acronyms have never been defined in the abstract 

Introduction:

The introduction is very well written and presents a solid background and rational to justify the current study. I appreciated the clarification of the difference between refractory vs resistant. I appreciated the background on what is already known and demonstrated on the dysfunction of endocannabinoid system among patients with migraine, also citing studies conducted by this group of research. 

A minor suggestion is to be specific already in the aim of the study in clarifying that the aim is to evaluate ECs dysfunction in patients with resistant migraine patients. 

Methods: 

-              Line 313: “and then returned to”

Very thorough method section, reproducible and solid in its statistical analysis and account for outliers and non-normally distributed data.

Where these patients taking any antidepressive, antipsychotic medications or mood stabilizers? This could be taken into consideration and adjust for in the statistical comparison between the groups 

Results: 

Very nicely summarized, tables and figures are pertinent and accurately enhance the understanding of the result section. 

Discussion: 

Very well written and summarized. 

Minor point: line 269 “it should be noted that”

Limitation: 

All the limitations identified by the authors are accurate. I would add that this specific population was only resistant migraine patients. As no control group of treatable migraine patient was considered, we do not know if these findings are what make migraine in these patients more difficult to be treated or if this is a characteristic of all the patients suffering from migraine. Briefly, the lack of a control group is a limitation. 

Author Response

We thank the reviewer for the guidance. We have done our best to act on the requests.

(Reviewer 1)

Thank you for inviting me to review the manuscript entitled “Peripheral endocannabinoid components and lipid plasma levels in patients with resistant migraine and co-morbid personality and psychopathological disorders: A cross-sectional study”. 

This is a cross sectional study conducted on patients with resistant migraine. Briefly, the authors characterized this cohort according to their psychological disorders and quantified the plasma levels of PEA, DAGLalfa and CB receptor gene expression. 

The study has several strengths below identified. I could only point at one major shortcoming, which can be easily acknowledge in the limitation section. Also, other minor corrections are below highlighted. 

R: We thank the reviewer for appreciating our work and for giving us his valuable suggestion.

Abstract:

-              Line 19 -> “psychopathologies” would be better replaced by “psychological disorders”. Also, it would be better to be consistent throughout the abstract and the manuscript. For example, in line 24, these conditions are named as “psychopathological disorders”, while in line 21 they are referred to as “psychiatric disorders”. 

 R: We changed psychopathological into psychological consistently across the manuscript.

-              Line 28: PEA and CB1 as acronyms have never been defined in the abstract 

R: Done

Introduction:

The introduction is very well written and presents a solid background and rational to justify the current study. I appreciated the clarification of the difference between refractory vs resistant. I appreciated the background on what is already known and demonstrated on the dysfunction of endocannabinoid system among patients with migraine, also citing studies conducted by this group of research. 

A minor suggestion is to be specific already in the aim of the study in clarifying that the aim is to evaluate ECs dysfunction in patients with resistant migraine patients. 

R: The purpose was better specified.

Methods: 

-              Line 313: “and then returned to”

R: The typo has been corrected

Very thorough method section, reproducible and solid in its statistical analysis and account for outliers and non-normally distributed data.

Where these patients taking any antidepressive, antipsychotic medications or mood stabilizers? This could be taken into consideration and adjust for in the statistical comparison between the groups 

R: Thank you for this consideration. The number and the percentage of patients ongoing to antidepressant treatment has been added to the text and to tables. We have added the concomitant treatment with antidepressant medications, together with monthly migraine days as suggested by Reviewer 3, to the multivariate regression model. Our results were confirmed also after these corrections. We thank for this consideration making our findings more solid. Follow-up studies evaluating the outcome of treatment with monoclonal antibodies will better clarify this aspect.

Results: 

Very nicely summarized, tables and figures are pertinent and accurately enhance the understanding of the result section. 

Discussion: 

Very well written and summarized. 

Minor point: line 269 “it should be noted that”

R: This typo has been fixed.

Limitation: 

All the limitations identified by the authors are accurate. I would add that this specific population was only resistant migraine patients. As no control group of treatable migraine patient was considered, we do not know if these findings are what make migraine in these patients more difficult to be treated or if this is a characteristic of all the patients suffering from migraine. Briefly, the lack of a control group is a limitation. 

R: This issue was addressed in the discussion and included in the limitations.

Reviewer 2 Report

Comments and Suggestions for Authors

This is a very interesting and intriguing original research article titled: “Peripheral Endocannabinoid components and lipid plasma levels in patients with resistant migraine and co-morbid personality and psychopathological disorders: A cross-sectional study”. The overall impression of the article is excellent.      

Introduction explain the rational driving this research. The aim of the study is to evaluate whether ECs dysfunction may have a role in resistant migraine when associated with psychological disorders. (lines 73-79) The research primary outcome measure was the difference in CB1 receptor expression in the different groups studied (line395-400). The primary outcome should be stated at the introduction section.

The research methodology, laboratory procedures and statistical methods are reported thoroughly allowing for replication.

              The lack of differences in psychological characteristics between the PD and wPD (lines 91-92) means that the used questionnaires cannot detect the diagnosis of PD using the SCID-5-CV. On the other hand, the presence of significant differences in their psychological characteristics at the Mood Disorders group probably means that the selected questionnaires detected their psychopathology. Perhaps using another appropriate to detect PD questionnaire would be helpful so as to highlight the dife\fernces between these groups (PD/WPd).

Authors very accurately states that the comorbidity percentage was high, and no “pure” patient groups were studied (lines 285-286). Taking into consideration that Episodic Migraine and Chronic Migraine have different pathophysiologic paths, there are four groups of patients in each type of migraine ie: High Frequency Episodic Migraine PD/WPD - Chronic Migraine PD/wPD and HFEM MD/wMD – CM MD/wMD. Please comment on the use of the Migraine type (Episodic/Chronic) as a confounder additionally to age and sex.

Typo at Line 313: “They were instructed to record headache characteristic in an ad hoc diary for one month (run-in period) and the returned to the center to receive the first dose of mAB (Visit 1).” 

Lines 231-235: “Furthermore, if we consider that most of the disorders in the PD group referred to Cluster C, being characterized by stress-reactive and control maladaptive personalities, probably this inference could be further supported by the fact that difficult-to-treat migraine patients probably experienced. Numerous physiological and behavioral changes caused by chronic stress include social deficits, decreased endocrine function, and an increased chance of developing psychiatric illness”  The full stop at line 234 should be removed.

Author Response

We thank the reviewer for the guidance. We have done our best to act on he/his  requests.

(Reviewer 2

This is a very interesting and intriguing original research article titled: “Peripheral Endocannabinoid components and lipid plasma levels in patients with resistant migraine and co-morbid personality and psychopathological disorders: A cross-sectional study”. The overall impression of the article is excellent.     

Introduction explain the rational driving this research. The aim of the study is to evaluate whether ECs dysfunction may have a role in resistant migraine when associated with psychological disorders. (lines 73-79) The research primary outcome measure was the difference in CB1 receptor expression in the different groups studied (line395-400). The primary outcome should be stated at the introduction section.

R: As requested, we stated the primary outcome of the study already in the introduction.

The research methodology, laboratory procedures and statistical methods are reported thoroughly allowing for replication.

              The lack of differences in psychological characteristics between the PD and wPD (lines 91-92) means that the used questionnaires cannot detect the diagnosis of PD using the SCID-5-CV. On the other hand, the presence of significant differences in their psychological characteristics at the Mood Disorders group probably means that the selected questionnaires detected their psychopathology. Perhaps using another appropriate to detect PD questionnaire would be helpful so as to highlight the dife\fernces between these groups (PD/WPd).

R: We agree on this point. However, it should be noted that we used only self-report questionnaires to assess medication dependence, depressive and anxiety symptoms and quality of life reported at the specific time they were completed. Therefore, it is to be expected that they are not sensitive to more stable diagnoses defined according to DSM-5 criteria. However, we totally agree with the reviewer who suggests the use of more sensitive questionnaires that can detect significant differences between groups. Since in the context of the present study we were interested in detecting ECs differences between groups, we reduced the repertoire of questionnaires used.

Authors very accurately states that the comorbidity percentage was high, and no “pure” patient groups were studied (lines 285-286). Taking into consideration that Episodic Migraine and Chronic Migraine have different pathophysiologic paths, there are four groups of patients in each type of migraine ie: High Frequency Episodic Migraine PD/WPD - Chronic Migraine PD/wPD and HFEM MD/wMD – CM MD/wMD. Please comment on the use of the Migraine type (Episodic/Chronic) as a confounder additionally to age and sex.

R: Thanks for the consideration that we agree with. As reported to Reviewer 3, we have added monthly migraine days (together with sex, age, and ongoing antidepressant medications) to the multivariate regression model. Our results were confirmed also after these corrections. We thank for this consideration making our findings more solid.

Typo at Line 313: “They were instructed to record headache characteristic in an ad hoc diary for one month (run-in period) and the returned to the center to receive the first dose of mAB (Visit 1).”

R: We fixed this typo.

Lines 231-235: “Furthermore, if we consider that most of the disorders in the PD group referred to Cluster C, being characterized by stress-reactive and control maladaptive personalities, probably this inference could be further supported by the fact that difficult-to-treat migraine patients probably experienced. Numerous physiological and behavioral changes caused by chronic stress include social deficits, decreased endocrine function, and an increased chance of developing psychiatric illness”  The full stop at line 234 should be removed.

R: As requested, we removed the full stop and merged the two sentences.

Reviewer 3 Report

Comments and Suggestions for Authors

The first time that abbreviations appear, they must be defined, regardless of whether it is done in the abstract. For example, PD and MD at the beginning of the Results section.

I, initially, did not understand what clusters A, B and C are. They appear for the first time in Results, without being defined, and then appear in Table 1. Then they are already defined in Methods. It would be appropriate to do it in table 1. Then all this is not used, since the only thing that is considered is PD and MD.

I hardly understood the philosophy of this paper. Comparing PD with non-PD, and MD with non-MD, and, of the group, only 75% had CM. In conclusion, they show differences in one parameter between PD and non-PD and in two parameters between MD and non-MD. The question is, what is the difference between PDs and non-PDs who do not have persistent migraine? What is the difference between MDs and non-MDs who do not have persistent migraine?

This is unclear and speculative work. I do not criticize the hypothesis, which seems good to me, but which, in my opinion, is not proven.

It cannot be published under these conditions

Author Response

We thankthe reviewer for the guidance. We have done our best to act on his/her requests, and to help to evaluate our revision, we provide a point-by-point discussion of how we addressed each one.

(Reviewer 3)

 The first time that abbreviations appear, they must be defined, regardless of whether it is done in the abstract. For example, PD and MD at the beginning of the Results section.

R: we reviewed the entire paper and checked for the definition of all abbreviations.

I, initially, did not understand what clusters A, B and C are. They appear for the first time in Results, without being defined, and then appear in Table 1. Then they are already defined in Methods. It would be appropriate to do it in table 1. Then all this is not used, since the only thing that is considered is PD and MD.

R: Cluster classification was reported to provide a more accurate description of the participants. As requested, in Table 1 we have now explained the meaning of each cluster of personality disorders and underlined the cluster names to differentiate them from their component disorders.

I hardly understood the philosophy of this paper. Comparing PD with non-PD, and MD with non-MD, and, of the group, only 75% had CM. In conclusion, they show differences in one parameter between PD and non-PD and in two parameters between MD and non-MD. The question is, what is the difference between PDs and non-PDs who do not have persistent migraine? What is the difference between MDs and non-MDs who do not have persistent migraine?

R: In the present paper, we focused on patients with a resistant migraine phenotype. According to the most up-to-date definition (Sacco et al. 2020), patients with resistant migraine should suffer a minimum of 8 monthly migraine days (MMDs). Thus, our population comprises high-frequency episodic migraine (8 to 14 MMDs) as well as chronic migraine (more than 15 MMDs) patients. We believe that our sample is indeed representative of the resistant migraine phenotype and of migraineurs eligible to receive anti-CGRP monoclonal antibody prophylaxis therapy.

Regarding the analysis, the distribution of chronic migraine is well balanced between PD and wPD, and MD and wMD groups. In addition, to account for this, we now added migraine frequency (namely monthly migraine days) in the multivariate regression models, confirming our previous results.

This is unclear and speculative work. I do not criticize the hypothesis, which seems good to me, but which, in my opinion, is not proven.

R: We are sorry for this consideration and we can understand it. We believe that resistant migraine represents a very difficult population of migraneurs that deserves to be fully studied by expanding the factors that may be involved in its pathophysiology, especially in case of personality and/or mood disorders. Such an evaluation is essential in order to differentiate this complex group of patients into different phenotypic and/or endotypic subtypes. Most studies to date have assessed ECs and related-lipids in migraine sufferers without making any distinctions based on psychological comorbidities. We believe that our study's attempt to shed light on this issue is intriguing and is extremely important. This is because it sheds light on an association that has been little explored so far and opens the way to the development of new therapeutic approaches for a particularly challenging migraine population. We are aware that our study presents many limitations, as we have already explained in the paper itself. However, we believe that it could represent a first step in order to identify the best candidate to monoclonal antibodies treatment.

Round 2

Reviewer 3 Report

Comments and Suggestions for Authors

There are 51 patients. 22 have PD and MD comorbidity. 29 patients have no comorbidity. How many patients are only PD? How many patients are MD only? think these data are relevant.

I don't know if it is possible to analyze separately those with comorbidity, those with only PD, and those with only MD.

It is not clear to me why they do not have pure groups, as indicated in the "Limitations of the study" section. If it is known that 22 have comorbidity, there are 29 that do not have it. I don't know if it's not well defined, but I haven't been able to find the tearjerker.

I think it would be convenient to have a table in a section of supplementary material where the patients could be seen individually, with all the characteristics. As presented, this study seems to me irreproducible.

I think the hypothesis is correct, but I also think it is necessary to provide more information before publication.

Before publishing this paper, these aspects should be clarified.

Author Response

Dear reviewer,

we have addressed the comment, as suggested.

In particular, we thank for this consideration. We are aware of this limitation and as suggested we have included the stratification of the population between the various groups in Supplementary Table 1.

However, due to the limited number of patients in the various subgroups, we are currently unable to make comparisons based solely on the interaction between these two variables. This will certainly be an aspect to be considered in future studies with a larger sample size.

The term “pure” has been replaced by "distinct".